# Lymphoproliferation Impairment and Oxidative Stress in Blood Cells from Early Parkinson’s Disease Patients

**DOI:** 10.3390/ijms20030771

**Published:** 2019-02-12

**Authors:** Carmen Vida, Hikaru Kobayashi, Antonio Garrido, Irene Martínez de Toda, Eva Carro, José Antonio Molina, Mónica De la Fuente

**Affiliations:** 1Department of Genetics, Physiology and Microbiology, Faculty of Biology, Complutense University of Madrid, 28040 Madrid, Spain; mvida@ucm.es (C.V.); hkgarcia@ucm.es (H.K.); antoniogarridobio@gmail.com (A.G); irene_mc90@hotmail.com (I.M.d.T.); 2Institute of Biomedical Research Hospital 12 Octubre (imas12), 28041 Madrid, Spain; carroeva@h12o.es (E.C.); cvillaiza@telefonica.net (J.A.M); 3Centro de Investigación Biomédica en Red sobre Enfermedades Neurodegenerativas (CIBERNED), 28040 Madrid, Spain

**Keywords:** biomarkers, blood cells, immune functions, immunosenescence, lymphoproliferation, oxidative damage, oxidative stress, Parkinson’s Disease

## Abstract

In Parkinson’s Disease (PD), the peripheral changes in the functional capacity and redox state of immune cells has been scarcely investigated, especially in the early PD stages. Aging is a risk factor for PD, and the age-related impairment of the immune system, based on a chronic-oxidative stress situation, is involved in the rate of aging. We analyzed several functions in isolated peripheral blood neutrophils and mononuclear cells from PD stage 2 patients, and compared the results to those in healthy elderly and adult controls. Several oxidative stress and damage parameters were studied in whole blood cells. The results showed an impairment of the lymphoproliferative response in stimulated conditions in the PD patients compared with age-matched controls, who also showed typical immunosenescence in comparison with adult individuals. Higher oxidative stress and damage were observed in whole blood cells from PD patients (lower glutathione peroxidase activity, and higher oxidized glutathione and malondialdehyde contents). Our results suggest an accelerated immunosenescence in PD stage 2, and that several of the parameters studied could be appropriate peripheral biomarkers in the early stages of PD.

## 1. Introduction

Parkinson’s disease (PD) is a chronic progressive, age-related, neurodegenerative disorder that is clinically characterized by the presence of predominantly motor symptomatology (e.g., bradykinesia, rest tremor, etc.), together with a diversity of non-motor symptoms (e.g., pain, paresthesia, depression, etc.) [1,2]. Neuropathologically, PD is characterized by the degeneration of neurons in the substantia nigra pars compacta (SNpc), which results in a loss of dopamine (DA) [3], as well as by the presence of intraneuronal Lewy bodies, which are neuronal inclusions of aggregated α-synuclein and other ubiquitinated proteins [4]. Although the exact mechanism of development and progression of PD pathology is still unknown, epidemiological studies have revealed that PD is a multifactorial disease with a complex combination of genetic background and environmental factors, aggravated by aging, which is, to date, the most common risk factor for neurodegenerative diseases [5]. In this regard, PD is the second most prevalent neurological disease, affecting at least 1% of the population over 50 years of age, and increases to 1–3% in subjects over 80 years of age [6,7]. Although medications alleviate PD symptoms, there is no cure nor are there clearly effective disease-modifying treatments [8]. There is no definitive biomarker or imaging test for PD and the diagnosis is determined from clinical observations and confirmed only by postmortem neuropathological analysis [8]. However, considerable efforts have been invested in the identification of peripheral biomarkers that can diagnose PD in the early stages of clinical development so that pharmacological interventions can be more effectively provided [8,9]. Unfortunately, by the time PD symptoms are clinically evident, 50–60% of neurons in the SNpc have been lost and DA levels in the striatum are depleted [8,10]. Since conducting studies in the early stages 1 and 2 of PD is difficult, most studies are performed using a mixed population with patients from different stages of PD (stages 2 to 5), which complicated the identification of early biomarkers of the onset and progression of PD.

Aging is accompanied by a decline in the nervous and immune systems, contributing to the deterioration of homeostasis and health [11,12]. For this reason, PD can be understood in the context of aging. Thus, the age-related decline of the immune system, which is called immunosenescence, should be considered in the development of this pathology. Most experimental data on immune changes with aging show a decline in innate and adaptive immune functions, which negatively affect the health of older adults, increasing morbidity and mortality [11,12]. Thus, there is an age-related increase in several leukocyte functions (e.g., adherence capacity and basal lymphoproliferation), whereas other functions decrease with age (e.g., phagocytosis, chemotaxis or stimulated proliferation) [11,12,13]. Given the bidirectional communication between the immune and the nervous systems [11,12], it is possible that the immunological variations contribute to the progressive neuronal dysfunction and PD pathology. Several studies supported that PD is also accompanied by a systemic disturbance, reflecting the damage in the brain. Although many contradictory results have been published, several studies have demonstrated an alteration in both innate and adaptive immune functions [14,15,16], deregulation of the cytokine production, as well as significant changes in subpopulation distributions of peripheral immune cells in PD patients compared with healthy elderly subjects [17,18,19,20,21]. This evidence suggests the possibility of developing potential therapies targeting the inflammatory and immune responses, whereas functional profiling of peripheral immune cells could provide important PD biomarkers [16]. It is crucial to identify and elucidate the disparity between the changes in innate and adaptive immunity throughout the different stages of PD, as well as to evaluate the impact of immunosenescence on the onset and progression of this disease, which remains incompletely clarified [17].

A chronic oxidative stress situation (a progressive imbalance between higher endogenous levels of oxidants and lower antioxidant defenses), which leads to the oxidative damage of cell molecules (lipids, proteins and DNA), is the basis of aging and several age-related diseases [11]. In this regard, oxidative stress has been implicated as a major mechanism in the pathogenesis and progression of PD [22,23,24]. This has been supported by several studies demonstrating an increase in oxidative stress and damage in several brain regions of PD patients [25,26,27,28,29]. Lower reduced glutathione (GSH) [27], altered activities of antioxidant enzymes, such as glutathione peroxidase (GPx) and glutathione reductase (GR) [27,28], or increased oxidative damage of lipids, proteins, and DNA [27,29,30,31], have been described in the brains of advanced-stages PD patients. Other studies revealed altered oxidative stress markers in cerebrospinal fluid (CSF), serum, plasma, or red blood cells (RBCs) of PD patients [23,32,33,34,35]. However, the changes in the redox stage of peripheral immune cells from PD patients in early stages of the disease are still unclear.

Finally, whole blood (WB) cells, and particularly circulating leukocytes, reflect the major physiological changes in various body organs and systems [36]. Therefore, the assessment of several immune markers in peripheral WB represents an accessible and useful method of studying and monitoring the immune response in PD, as well as for identifying new biomarkers that could be predictive of the disease status and measures of disease severity and progression. In this context, recent studies proposed that measurement of the immune function profile in different types of peripheral leukocytes, together with the evaluation of their oxidative-inflammatory state, could be useful early peripheral markers of the progression of neurodegenerative diseases [36,37]. However, this kind of study in the early stages of human PD is a subject that has scarcely been explored.

With the above in mind, the aim of the present work was to study the changes in several immune functions, and oxidative stress and damage parameters in different types of human blood immune cells at an early stage of PD pathology. As such, we performed immune functions assays on both isolated peripheral blood polymorphonuclear (mainly neutrophils) and mononuclear (mainly lymphocytes) leukocytes. The oxidative stress and damage parameters were assessed in WB cells, from patients diagnosed with PD stage 2 and age-matched controls (elderly healthy subjects), as well as in adult controls, to evaluate the differences due to aging.

## 2. Results

### 2.1. Impairment of Adaptive Immune Functions Presented in PD Stage 2 Patients

Several immune functions parameters were studied in isolated human blood neutrophils and mononuclear leukocytes from PD stage 2 patients, as well as from adult and elderly healthy subjects. The results are shown in Figure 1 and Figure 2.

In general, all the immune parameters analyzed in this work (adherence, chemotaxis, and phagocytosis of neutrophils; the antitumor cytotoxic activity of natural killer (NK) cells; adherence, chemotaxis, and both basal and stimulated proliferative response of lymphocytes) deteriorate with aging. However, a differential pattern of the innate and adaptive immune responses was observed in PD stage 2 patients, which showed an impairment of the adaptive immune functions but not in the innate response compared with age-matched controls.

In the early stage of PD pathology, and unlike what is observed with aging, no significant differences were observed between PD patients and healthy controls of the same chronological age in any of the innate immune functions analyzed, such as adherence (Figure 1A,B), chemotaxis (Figure 1C,D), phagocytosis (Figure 1E,F), and NK cytotoxic activity (Figure 1G). However, with aging, elderly healthy subjects showed statistically significant lower values in both neutrophil (*p* < 0.001; Figure 1C) and lymphocyte (*p* < 0.05; Figure 1D) chemotaxis and the phagocytic index (*p* < 0.001; Figure 1E), as well as in NK cell cytotoxic activity (*p* < 0.01; Figure 1G), in comparison with healthy adult subjects. In contrast, higher values of neutrophil and lymphocyte adherence (*p* < 0.05; Figure 1A,B) were observed in elderly healthy subjects compared with adult controls.

In contrast, with regards to adaptive immune functions (lymphoproliferation in basal and stimulated conditions), these functions were further affected in PD patients compared with their age-matched controls. The levels of both phytohaemagglutinin (PHA)- (*p* < 0.001; Figure 2A) and lipopolysaccharide (LPS)- (*p* < 0.05; Figure 2B) lymphoproliferation were significantly lower in PD patients than those observed in elderly controls. Similar results were observed in the lymphoproliferation capacity (% stimulation) (*p* < 0.05; Figure 2C,D). However, no significant differences were observed in basal lymproliferation between both experimental groups (3850 ± 2036 and 4009 ± 1973 c.p.m., PD stage 2 patients vs. elderly control, respectively). With aging, elderly subjects showed significantly lower values of both PHA- and LPS-stimulated lymphoproliferation (*p* < 0.001; Figure 2) compared with those observed in adult subjects. In contrast, higher values of basal lymphoproliferation (*p* < 0.01) were observed in elderly healthy subjects in comparison with adult controls (4009 ± 1973 and 1363 ± 452 c.p.m. in elderly vs. adult subjects, respectively).

### 2.2. Oxidative Stress and Lipid Damage in Whole Blood cells from PD Stage 2 Patients

Several oxidative stress and damage parameters were analyzed in isolated whole blood (WB) cells (containing both total RBCs and leukocyte populations) from PD stage 2 patients, as well as from adult and elderly healthy subjects. The results are shown in Figure 3.

In their WB cells, PD patients exhibited significantly lower and higher glutathione peroxidase (GPx) (*p* < 0.01; Figure 3A) and glutathione reductase (GR) (*p* < 0.05; Figure 3B) activities, respectively, as well as higher oxidized glutathione (GSSG) (*p* < 0.05; Figure 3D) and malondyaldehide (MDA) (*p* < 0.001; Figure 3F) contents than elderly subjects. However, no statistically-significant differences were observed in GSH concentrations and GSSG/GSH ratios between PD and elderly individuals. Regarding aging, higher values of oxidative stress and damage were observed in WB cells of elderly subjects than in those of adults. Thus, GPx, GR, and GSH values were lower (*p* < 0.001, *p*< 0.01 and *p* < 0.01, respectively; Figure 3A–C) and GSSG/GSH ratios (*p* < 0.05; Figure 3E), as well as both GSSG (*p* < 0.001; Figure 3D) and MDA (*p* < 0.05; Figure 3F) levels were higher than those in adult subjects.

The study of the effect of treatment with antiparkinsonian medications, such as levodopa (L-DOPA), is not the focus of the present work. However, since five of the enrolled (highlighted in green in Figure 1, Figure 2 and Figure 3) patients were taking L-DOPA for more than six months, and due to several studies having provided some evidence that levodopa treatment could affect the redox state and oxidative damage at peripheral levels (e.g., blood, plasma or peripheral blood mononuclear cells) [33,34,38], we examined if L-DOPA treatment could affect to the oxidative state of these “medicated PD patients” in comparison to “non-medicated PD patients”. We observed that, in all the oxidative stress and damage parameters analyzed, in general, the five “medicated” PD stage 2 patients (see highlighted green points in Figure 3) showed similar values to the observed “non-medicated” PD subjects (Table 1). Similar results were also recorded for all the immune functions parameters analyzed; no statistical differences were observed in these parameters between both groups of PD patients (see highlighted green points Figure 1 and Figure 2; Table 1). Therefore, our results reveal that L-DOPA therapy did not significantly influence the measured immune functions and oxidative stress in peripheral blood cells of PD stage 2 patients.

## 3. Discussion

The state of the peripheral immune system in PD is increasingly being established as a hallmark of the disease [17,39]. Although several studies have reported alteration of both innate and adaptive immunity at the moderate and advance stages of PD, many contradictory results have been reported. Some authors point to an overactive immune function, whereas others postulate the existence of a general immune deterioration in PD compared to healthy individuals of the same age [16,20,40,41]. Most of these studies focused on the alteration of the adaptive immune response in which T cells play a key role in the neurodegenerative process of PD [15,17,42]. However, little is known about these changes in both innate and adaptive immunity in the early stages of PD. To the best of our knowledge, this is the first study that demonstrated a differential pattern of the peripheral immune responses in early stage 2 PD patients, which demostrates an early impairment of both PHA- and LPS-stimulated lymphoproliferative responses, but not in their innate functions, compared with elderly healthy subjects of the same chronological age. This suggests that PHA- and LPS-lymphoproliferation parameters, characteristic of adaptive immunity, could be useful early peripheral biomarkers of the appearance of PD. The decrease in the proliferative response of lymphocytes to mitogens (e.g., LPS and PHA), one of the most evident age-related changes of the immune system, has been associated with a higher risk of morbidity and mortality [13,43], and together with other parameters, defines the immune risk phenotype in humans [44,45]. In regards to aging, PD stage 2 patients and elderly controls showed the typical immunosenescence in comparison with adult individuals (lower chemotaxis, phagocytosis, NK activity, and stimulated-lymphoproliferation; higher adherence and basal lymphoproliferation) [11,13,43]. However, the age-related impairment observed in both PHA- and LPS-lymphoproliferative responses were more exacerbated in early PD patients than in elderly controls, suggesting that these subjects suffer an accelerated impairment of the acquired immunity at the early stage 2 of PD. In this regard, although strong clinical data support that peripheral inflammation appears to be an early event in the development of PD, the mechanism through which PD patients suffer an accelerated impairment of the peripheral acquired immunity in early stages of PD are not yet clear. The growing evidence for such possible mechanisms includes a decreased number of peripheral blood lymphocytes or alteration of cytokine production by peripheral blood cells [20,46,47,48]. For instance, a significant reduction in the B and T cells subpopulation repertoire, together with increased levels of pro-inflammatory cytokines, were observed in peripheral blood cells or serum of PD patients, as well as in animal models for PD [17,18,19,20,21,42,46].

With respect to PHA- and LPS-lymphoproliferative responses, our results agree with other studies that supported an impaired mitogenic response in peripheral immune cells from subjects with PD [40,49,50]. A marked decreased proliferation in response to a wide range of mitogens (LPS, PHA, or Concanavalin A) was observed in splenocytes and peripheral blood mononuclear cells (PBMNs) in mice and patients with PD [40,49,50]. Other authors pointed to a hyperactivity of T cells in response to LPS activation [20,41]. This lack of consensus could be due to the complex modulation of the adaptive immune response, as well as to the different stages of the disease on which these studies focused (mainly in a mixed population of subjects at different stages of PD). In the context of aging, age-related alterations in adaptive immunity have been extensively studied and T cells are extremely sensitive to immunosenescence [11,12]. T-lymphocyte proliferative responsiveness to antigens or mitogens decreases with age, which is due, at least in part, to a decreased expression of the costimulatory molecule CD28 and a lower production of T cell growth factor (IL-2) [51]. Considering these findings, the impairment of PHA- and LPS-lymphoproliferation observed in PD stage 2 patients could be the result of lower T cells count or variations in the IL-2 producing T cell population after mitogen stimulation, which may be due to a lack of co-stimulatory signals. Several studies have found differences between circulating leukocyte populations in PD patients compared with controls, postulating that other authors have found a significantly reduced secretion of IL-2 by PBMCs from PD patients following mitogen stimulation [40]. However, it is also important to highlight that, unlike observed with aging, no differences were observed in basal lymproliferation between PD stage 2 patients and the elderly control. Although we have not found any study on isolated PBMNs from PD patients, an increased basal proliferation was reported in lymphoblasts of moderate and severe PD subjects [52]. Basal lymphoproliferation provides information about the immune system overactivation, and it has recently been proposed as a useful age-related marker [43]. Therefore, our results suggest that the immune system of PD patients is not overactivated during the early stages of the disease, which could be attributed to the good maintenance of the basal proliferative response, potentially as a compensatory mechanism for the lack of effectiveness of lymphoproliferation in response to stimulus.

Our results revealed that PD stage 2 patients did not show an alteration in their peripheral innate immune responses, displaying similar neutrophils and lymphocyte adherence and chemotaxis, phagocytic capacity, as well as NK cytotoxic activity to those observed in age-matched controls. Although there is no study that, at early stages of PD, has evaluated similar innate immune functions as those analyzed in this work, an alteration in the innate response has been observed in peripheral immune cells from mild and severe PD patients [15,17,41,53]. Thus, in contrast to our results, a reduction in the phagocytic activity of monocytes and an altered activation of NK cells were observed in these PD patients [41,54]. PD patients also showed an increased percentage of NK cells, which correlates with PD severity scores; however, in agreement with our results, no differences in the NK activity were observed in these PD patients in relation to age-match controls [53,54,55]. All these modifications to NK cell physiology suggest that they might play a role in the pathogenesis of PD. Since we cannot explain the possible mechanisms that PD patients use to preserve their innate functions at early stages of PD, further studies are needed to fully understand these mechanisms.

Chronic oxidative stress plays a key role in aging, as well as in the pathogenesis of PD [11,34,56]. Since the maintenance of a good redox state and a low accumulation of cellular damage are important for preserving immune function and achieving extreme old age [11,51], it is possible that the early deterioration of the stimulated proliferative response observed in PD stage 2 patients could be mediated by an increase in oxidative stress and damage affecting leukocytes. Higher levels of oxidative stress and lipid damage were observed in WB cells of PD patients in relation to age-match controls, identified as altered antioxidant GPx and GR activities and higher GSSG and MDA contents. Our results agree with those of other studies in which elevated production of oxidative compounds, increased markers of lipid peroxidation (e.g., MDA) damage, or the altered levels of antioxidant enzymes and compounds were also found in peripheral PBMNs, RBCs, serum and plasma of PD patients and animal models of PD [23,32,33,34,35]. However, because some studies showed an increase in oxidative stress [23,33,57,58,59], whereas other authors reported no changes [60], the significance of peripheral oxidative stress in PD remains to be clarified. These conflicting results may be due to the different methods used to measure systemic oxidative stress, lack of analyzing confounding factors that could influence the oxidative markers, the relatively small cohort in most studies, and the different PD stages during which the studies were completed. Notably, most studies were performed during the mild and severe stages of PD, or in a mixed population of patients in different stages of PD, so it was not possible to analyze redox state differences between early and late stages of PD.

The maintenance of adequate levels of antioxidants is essential to prevent or even manage changes in the redox balance throughout aging and during the onset and progression of PD. We analyzed the glutathione cycle, which is one of the main intracellular mechanisms for maintaining redox homeostasis [61]. In WB cells, PD stage 2 patients showed higher GSSG levels than elderly controls. However, GSH content and GSSG/GSH ratio were similar between both groups. This indicates that, during the early stages of PD, WB cells do not suffer a great alteration in their redox state (measured as GSSG/GSH ratio) as these cells are able to maintain optimal GSH levels, perhaps as a compensatory mechanism to manage the increase in oxidant production. Our results agree with some studies conducted in red blood cells (RBCs) in which increased GSSG levels [23], but not in total glutathione and GSH contents [23,38,62], were found in PD patients (mixed population of stages 1–4). The RBCs GSH is correlated with central nervous system GSH state and PD severity [61]. Our study also revealed that PD patients showed a different pattern in the activities of the GPx and GR antioxidant enzymes in WB cells, showing lower and higher GPx and GR activities, respectively, in comparison to elderly controls. Different studies have reported contradictory results. PD patients showed low GPx activity in WB cells [63], whereas no changes in GPx and GR activities were found in RBCs or plasma [64,65]. GPx is an enzyme that couples the oxidation of GSH to the detoxification of hydroperoxides, whereas the importance of GR lies in its ability to maintain GSH in its reduced form [64]. The higher GR activity observed in WB cells from PD stage 2 patients could be explained as a result of the activation of a compensatory response to cope with the high levels of GSSG observed in these cells. The lower GPx activity observed in these individuals may result in an accumulation of peroxides, which could contribute to causing lipid oxidative damage in these cells. This could explain the fact that the PD patients also showed higher MDA levels in their WB cells in relation to elderly controls. Our finding is also supported by other studies, which reported that MDA content significantly increases in plasma, serum, blood leukocytes or RBCs of PD patients [60,66,67,68]. In our study, the increased levels of MDA in peripheral WB cells, together with the excessive GR activity observed in PD stage 2 patients, may indicate a systemic reaction related to chronic oxidative stress in the brain.

In this study, we analyzed different oxidative stress and damage parameters using samples of WB cells but not isolated blood leukocytes for several reasons: (1) this kind of sample better reproduces the in vivo conditions in which the immune cells are living; (2) WB samples are clinically more feasible, reproducible, cost effective, easy to implement, and apply compared to purified and isolated neutrophils and mononuclear blood leukocytes; (3) a previous study completed in our laboratory demonstrated a similar altered redox state and oxidative damage pattern in WB cells and isolated blood neutrophils from AD patients, suggesting that the estimation of these parameters in WB cells would be a useful biomarkers in the assessment of AD progression [36]; and (4) a recent study by our group (submitted for publishing) assessed WB cells from health men and women throughout aging, revealing increased oxidative stress and damage in elderly subjects in comparison to adults, suggesting that the measurement of these redox state parameters in WB cells could be good biomarkers of biological age. In this context, in relation to aging, our findings also demonstrate that several peripheral oxidative stress and damage markers increased in WB cells from both PD stage 2 patients and elderly controls in comparison with adult individuals (lower GPx and GR activities and GSH content; higher GSSG/GSH ratios and GSSG and MDA contents). However, since increased oxidative stress and damage were higher in WB cells of PD stage 2 patients than in elderly controls, this suggests that these parameters (GPx and GR activities and GSSG and MDA contents) could be used as possible early peripheral biomarkers of PD.

In conclusion, our study demonstrated a different pattern of altered immune response in early stage 2 of PD. Thus, PD patients showed an impairment of lymphoproliferation in stimulated conditions but not in the innate responses compared with age-matched controls. As occurs in aging, this deteriorated immune response could have an oxidative stress situation as their basis. Thus, our findings also revealed that this altered lymphoproliferation could be mediated by the higher oxidative stress and damage also observed in WB cells from PD stage 2 patients, which may reflect a systemic reaction probably related to a chronic oxidative stress situation in the brain. Therefore, our results support the idea that PD stage 2 patients show accelerated immunosenescence due to te lower lymphoproliferation and higher oxidative stress and damage that these PD patients displayed in comparison to elderly subjects of the same chronological age. Due to (1) WB cells being easy to obtain and analyze in the clinical setting, and (2) several immune function parameters being described as good markers of the rate of aging, the process used may help with the early identification of accelerated aging in human, and may offer opportunities for prevention of premature death and age-related diseases [13]. Our results suggest that the assessment of several immune functions parameters in isolated blood neutrophils and mononuclear cells, together with the analysis of oxidative stress and damage parameters in WB cells, could be useful peripherical biomarkers of the onset and progression of PD. Additional studies are required in peripheral blood immune cells, especially in the early stages of PD to identify other potential prodromal and preclinical biomarkers of PD, which are crucial for early diagnosis of PD. This can help with the development of effective therapeutics and in the early intervention with proven therapies, which may help slow the onset of symptoms and reduce their intensity for as long as possible.

## 4. Materials and Methods

### 4.1. Study Subjects

For this cross-sectional study, 45 PD stage 2 patients (women/men, 15/30; mean age ± SD: 67 ± 12 years), as well as 34 elderly (women/men, 24/10; mean age ± SD: 74 ± 11 years) and 20 adult (women/men, 10/10; mean age ± SD: 40 ± 8 years) healthy volunteers were recruited by the Neurology Department of the Hospital 12 Octubre of Madrid, Spain, and were tested by a standardized neuropsychological battery. Diagnosis of sporadic PD was made according to the U.K. Parkinson’s Disease Brain Bank criteria and classified according to Hoehn and Yahr (HY) Rating Scale [9,69]. Subjects with parkinsonian symptoms due to vascular parkinsonism, or medicine- or toxin-induced parkinsonism were excluded. Notably, in the PD stage 2 group, there were 40 “unmedicated PD patients” (diagnosed with PD but not yet started on PD treatment) and 5 “medicated PD patients” (treated with PD medications, such as L-DOPA for more than six months). All patients were subjected to a clinical survey and physical examination. Individuals presenting with any of liver, renal, or cardiac dysfunction; cancer; malabsorption; autoimmune diseases; or with current infectious conditions (because immune functions parameters and oxidative stress markers in peripheral blood may be altered in such conditions) were excluded from both PD and control groups. All experiments were performed in accordance with guidelines and regulations of the Declaration of Helsinki. All procedures were approved by the corresponding Research Ethic Committee of the Biomedical Research Institute Hospital 12 Octubre (code: ES280790001164). Written informed consent was obtained from all participants or representatives.

### 4.2. Collection of Peripheral Whole Blood Cells and Isolation of Blood Neutrophils and Lymphocytes Leukocytes

Human samples (10 mL) of peripheral blood were collected using vein puncture and sodium citrate-buffered Vacutainer tubes (BD Diagnostic, Madrid, Spain). Blood extraction was performed between 8:00 a.m. and 11:00 a.m. to avoid circadian variations in immune parameters. On the one hand, 8 mL of peripheral blood was used for isolation of both polymorphonuclear (mainly neutrophils) and mononuclear (mainly lymphocytes) leukocytes following a previously described method [70]. Thus, neutrophil and lymphocyte cells were isolated using 1.119 and 1.077 g/cm^3^ density Histopaque (Sigma-Aldrich, Madrid, Spain) separation, respectively. Collected cells were counted (95% of viability determined using trypan blue staining) and adjusted to the corresponding final concentrations for the development of the different assays of immune functions. The immune functions assays were performed with fresh cells. Samples of whole blood (WB) cells, which contain the total red blood cells (RBCs) together with the total leukocyte populations, were obtained following a previously described procedure [36]. Several aliquots were prepared for the determination of redox state parameters, which were stored at −80 °C until used.

### 4.3. Adherence Capacity Assay

For the measurement of the adherence capacity of human peripheral blood neutrophils and lymphocytes, we followed a method previously described [70] with some slight modifications [36]. This method mimics, in vitro, cellular adherence to endothelium in vivo, using adherence columns consisting of a Pasteur pipet with 50 mg of nylon fibers. The percentage of adherent neutrophils and lymphocytes, expressed as Adherence Index (A.I.), was calculated according to the equation: AI: (leukocytes/mm^3^ total – leukocytes/mm^3^ effluent)/(leukocytes/mm^3^ total).

### 4.4. Chemotaxis Assay

The induced mobility or chemotaxis of neutrophils, macrophages and lymphocytes was evaluated according to the method previously described [36,70]. Cell suspensions were deposited in the upper compartment of a Boyden chamber separated by a filter of polycarbonate (3 μm in diameter; MercK, Madrid, Spain). The number of cells (neutrophils and lymphocytes) that migrated toward the chemoattractant agent, formyl-Met-Leu-Phe (fMLP, 10^−8^ M, Sigma-Aldrich, Madrid, Spain), deposited in the lower compartment of the chamber, was counted in the lower face of the filter that separates the two compartments of the chamber. This number is expressed as Chemotaxis Index (C.I.).

### 4.5. Phagocytosis Assay

Phagocytosis of inert particles (latex beads, 1.1 μm means particle size, Sigma-Aldrich, Madrid, Spain) was assayed in phagocytes (isolated blood neutrophils) following a method previously described [36,70]. The number of particles ingested by 100 neutrophils was counted using an immersion objective (100×) and this is expressed as phagocytic index (P.I.), whereas the number of ingesting neutrophils per 100 neutrophils is expressed as phagocytic efficiency (%).

### 4.6. Natural Killer (NK) Cytotoxicity Assay

The natural killer (NK) cell cytotoxicity was evaluated following an enzymatic colorimetric assay (Cytotox 96 TM Promega, Boeringher Ingelheim, Biberach, Germany) based on the determination of lactate dehydrogenase (LDH) released by the cytolysis of targets cells (human K562 lymphoma cells), using tetrazolium salts [70]. The results are expressed as the percentage of tumor cells killed (% lysis).

### 4.7. Lymphoproliferation Assay

The proliferation capacity of lymphocytes was evaluated by a standard method previously described by our laboratory [70]. The assay was assessed in both basal and stimulated conditions using mitogens [phytohemagglutinin (PHA) and lipopolysaccharide (LPS), 1 μg/mL, respectively; Sigma-Aldrich, Madrid, Spain]. The results were calculated as ^3^H-thymidine uptake (c.p.m.) for basal and stimulated conditions, and are expressed as lymphoproliferation capacity (%) giving 100% to the c.p.m. in basal conditions.

### 4.8. Glutathione peroxidase (GPx) Activity Assay

The glutathione peroxidase (GPx) activity was determined according to the technique described by Lawrence and Burk [71] with several modifications [36]. The assays were performed with aliquots of frozen WB cells (200 μL, containing total RBCs and total leukocyte populations), which were resuspended with phosphate buffer saline 50 mM, pH 7.5 (300 μL), and centrifuged at 3200× *g* for 20 min at 4 °C. The total activity was determined using cumene hydroperoxide (Sigma-Aldrich, Madrid, Spain), which oxidized the glutathione regenerated by the addition of β-nicotinamide adenine dinucleotide phosphate in its reduced form (β-NADPH, Sigma-Aldrich, Madrid, Spain), in the presence of glutathione reductase (Sigma-Aldrich, Madrid, Spain). The reaction was spectrophotometrically followed by the decrease in the absorbance at 340 nm. The protein contents were evaluated following the bicinchoninic acid protein assay kit protocol (Sigma-Aldrich, Madrid, Spain), using serum albumin (BSA, Sigma-Aldrich, Madrid, Spain) as standard. The results are expressed as units of enzymatic activity per milligram of proteins (U GPx/mg protein).

### 4.9. Glutathione Reductase (GR) Activity Assay

The glutathione reductase (GR) activity was measured by the method described by Massey and Williams [72] with some modifications [36]. The assays were performed with aliquots of frozen WB cells (100 μL) that were resuspended with phosphate buffer saline 50 mM with ethylenediaminetetraacetic acid (EDTA; Sigma-Aldrich, Madrid, Spain) 66 mM, pH 7.4 (300 μL), and centrifuged at 3200× *g* for 20 min at 4 °C. The total activity was spectrophotometrically measured through the oxidation of NADPH at 340 nm. The protein contents of the same samples were evaluated following the previously described protocol. The results are expressed as units of enzymatic activity per milligram of proteins (U GR/mg protein).

### 4.10. Glutathione Content Assay

Both reduced glutathione (GSH), the main non-enzymatic reducing agent of the organism, and oxidized glutathione (GSSG) forms of glutathione were determined using a fluorimetric assay previously described [73], with several modifications previously performed by our laboratory [36]. This method is based on the GSSG and GSH capacity for reaction with o-phthalaldehyde (OPT, Sigma-Aldrich, Madrid, Spain), at pH 12 and pH 8, respectively, resulting in the formation of a fluorescent compound. The assay was evaluated in aliquots of frozen WB cells (50 μL), which were resuspended in phosphate buffer saline 0.1 M, pH 7.4 (200 μL) (Sigma-Aldrich, Madrid, Spain). In both GSH and GSSG measurements, the fluorescence emitted by each well was measured at 350 nm excitation and 420 nm emission. Protein content of the samples was determined following the previously described protocol. Results are expressed as nanomoles of GSH or GSSG per milligram of protein (nmol/mg protein). The GSSG/GSH ratio was calculated for each sample.

### 4.11. Lipid Peroxidation (MDA) Assay

Determination of malondialdehyde (MDA) levels was evaluated using the commercial MDA Assay Kit (Biovision, Milpitas, CA, USA), which measures the reaction of MDA with thiobarbituric acid (TBA) and the MDA-TBA adduct formation. To accomplish this, frozen human WB aliquots (100 µL) were resuspended in MDA lysis buffer (200 µL) with 0.1 mM butylated hydroxytoluene (BHT) (2 µL), sonicated, and then centrifuged at 13,000× *g* for 10 min. The supernatants (200 µL) from each sample were added to 600 µL of TBA and incubated in a water bath at 95 °C for 60 min. Samples were cooled on ice for 10 min, and n-butanol (300 µL, Sigma-Aldrich, St. Louis, MO, USA) were added to create an organic phase in which the MDA molecules were placed. Samples were centrifuged for 10 min at 13,000× *g* at room temperature and the supernatants (300 µL of upper organic phase) were collected and dispensed into a 96-well microplate for spectrophotometric measurement at 532 nm. MDA supplied in the kit was used as a standard, and MDA levels were determined by comparing the absorbance of samples with those of the standards. Protein content of the samples was determined following the previously described protocol. Results are expressed as nanomoles of MDA per mg of protein (nmol/mg protein).

### 4.12. Statistical Analysis

Statistical analysis was performed in SPSS IBM, version 21.0 (SPSS, Chicago, IL, USA). All tests were two-tailed, with a significance level of *p* ≤ 0.05. Data are presented as mean ± standard deviation (SD). Normality of the samples and homogeneity of the variances were checked using the Kolmogorov–Smirnov test and Levene test, respectively. Differences due to PD pathology and age were studied using the Student *t*-test or one-way analysis of variance (ANOVA) followed by post hoc tests analysis or the non-parametric Kruskal-Wallis test. The Tukey test was used for post-hoc comparisons when variances were homogeneous, whereas its counterpart analysis, Games-Howell, was used with unequal variances when they were heterogenous. Figures were built using GraphPad Prism 6 Software (LLC, San Diego, CA, USA).

## Figures and Tables

**Figure 1 ijms-20-00771-f001:**
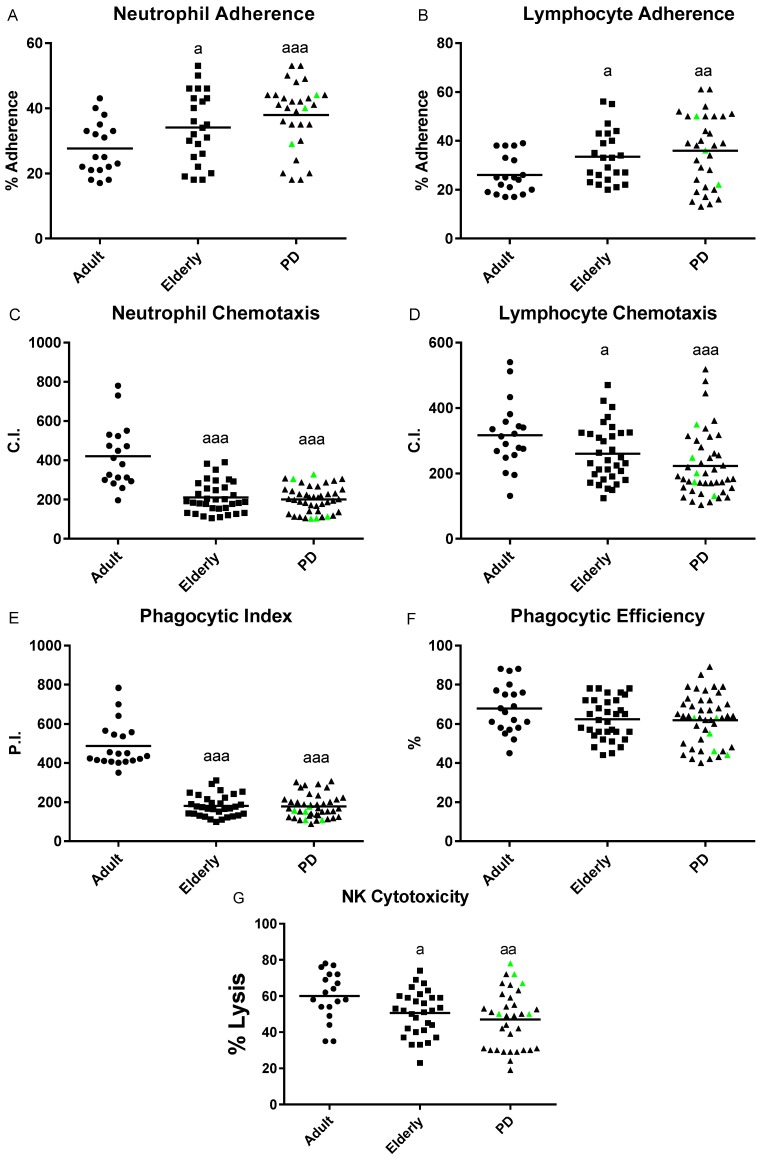
Immune function parameters in isolated peripheral blood neutrophils and mononuclear cells of Parkinson´s Disease (PD) stage 2 patients, and adult and elderly healthy controls. Chemotaxis index (C.I.) of (**A**) neutrophils and (**B**) lymphocytes; (**C**) phagocytic index (P.I.) and (**D**) phagocytic efficiency (%) of neutrophils and (**E**) natural killer (NK) cytotoxic activity (% lysis). Data are shown as the mean (horizontal bar) of 18–45 values corresponding to the numbers of subjects analyzed in each group (20 adults, 34 elderly, and 45 PD). Each point represents the mean of duplicate assays. The green points represent the values in five PD patients treated with levodopa. a: *p* < 0.05, aa: *p* < 0.01, and aaa: *p* < 0.001 with respect to the value in adult subjects.

**Figure 2 ijms-20-00771-f002:**
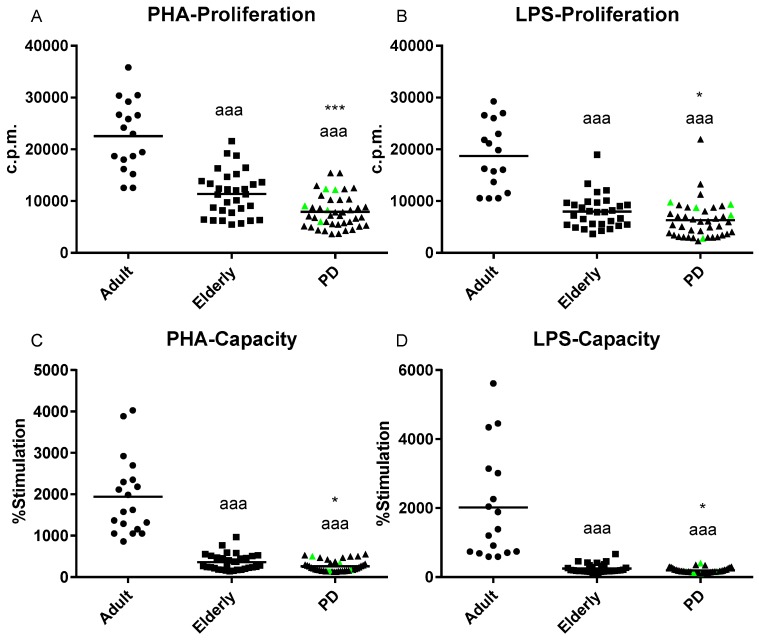
Stimulation of proliferation in response to phytohemagglutinin (PHA) and lipopolysaccharide (LPS) (1 μg/mL) in isolated peripheral blood mononuclear cells of Parkinson´s Disease (PD) stage 2 patients, as well as of adult and elderly healthy controls. Lymphoproliferation (48 h incubation) in counts per minute (c.p.m.) with the mitogens (**A**) PHA and (**B**) LPS. Lymphoproliferation capacity (%) in response to (**C**) PHA and (**D**) LPS providing 100% to the c.p.m. in basal conditions. Data are shown as the mean (horizontal bar) of 16–45 values corresponding to the number of subjects analyzed in each group (20 adults, 34 elderly, and 45 PD). Each point represents the mean of duplicate assays. The green points represent the values in five PD patients treated with levodopa. aaa: *p* < 0.001 with respect to the value in adult subjects; * *p* < 0.05 and *** *p* < 0.001 with respect to the value in elderly subjects.

**Figure 3 ijms-20-00771-f003:**
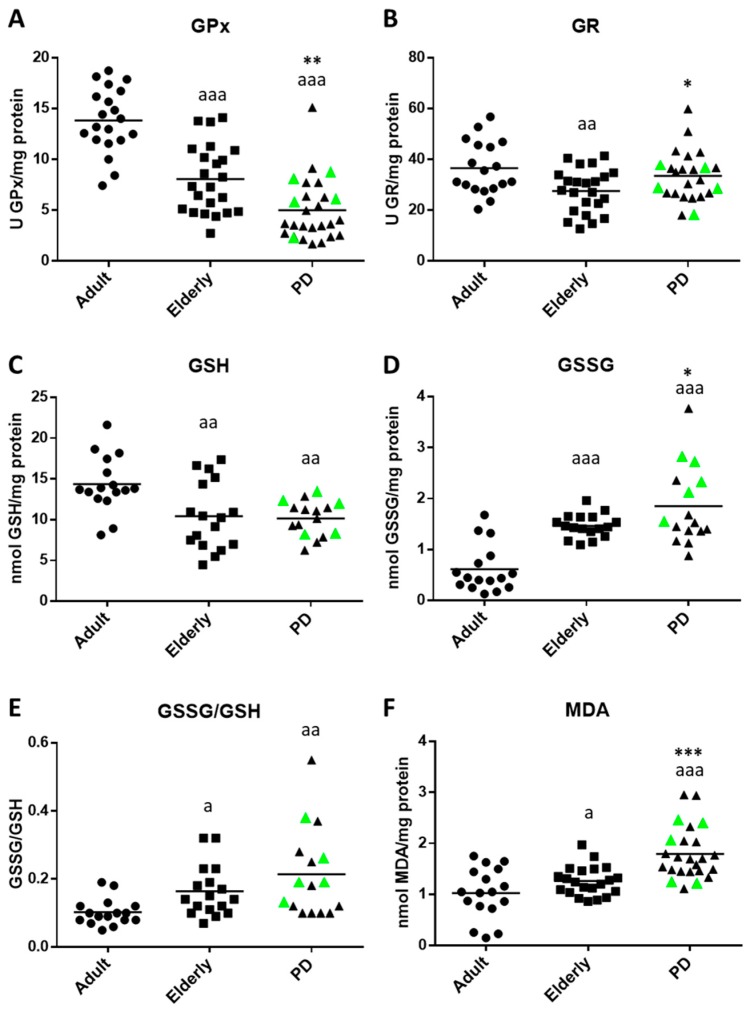
Oxidative stress and lipid peroxidation parameters in whole blood cells, containing both total red blood cells and leukocyte populations, of Parkinson´s Disease (PD) stage 2 patients, as well as of adult and elderly healthy subjects. (**A**) Glutathione peroxidase (GPx) and (**B**) glutathione reductase (GR) activities (U/mg protein); (**C**) intracellular reduced glutathione (GSH) concentrations and (**D**) oxidized glutathione (GSSG) contents (nmol/mg protein); (**E**) GSSG/GSH ratios; and (**F**) intracellular malondialdehyde (MDA) contents (nmol/mg protein). Data are shown as the mean (horizontal bar) of 16–23 values corresponding to the number of subjects analyzed in each group (20 adult, 23 elderly, 26 PD). Each value is the mean of duplicate assays. The green points represent the values in five PD patients treated with levodopa. a: *p* < 0.05, aa: *p* < 0.01, and aaa: *p* < 0.001 with respect to the value in adult subjects; ** p* < 0.05, *** p* < 0.01, and **** p*
*<* 0.001 with respect to the value in elderly subjects.

**Table 1 ijms-20-00771-t001:** Immune function parameters in isolated peripheral blood neutrophils (N) and mononuclear cells (L), and oxidative stress and lipid peroxidation parameters in whole blood cells from Parkinson’s Disease (PD) stage 2 patients, non-medicated and medicated with levodopa therapy for more than six months.

PD Stage 2 Patients	Non-Medicated	Medicated
**Immune Functions Parameters**	[24,25,26,27,28,29,30,31,32,33,34,35,36,37,38,39,40]	[3,4,5]
N adherence (%)	38± 10	35 ± 9
L adherence (%)	36 ± 14	34 ± 16
N chemotaxis (C.I.)	201 ± 10	218 ± 30
L chemotaxis (C.I.)	223 ± 15	213 ± 34
Phagocytosis index (P.I.)	177 ± 11	189 ± 24
Phagocytosis efficiency (%)	61 ± 4	56 ± 10
NK cytotoxic activity (% lysis)	47 ± 15	58 ± 19
Basal lymphoproliferation (cpm)	3827 ± 306	3931 ± 354
PHA-lymphoproliferation (cpm)	7920 ± 481	8090 ± 398
LPS-lymphoproliferation (cpm)	6306 ± 569	6629 ± 639
PHA-stimulated (%)	259 ± 21	284 ± 32
LPS-stimulated (%)	192 ± 18	203 ± 22
**Oxidative Stress Parameters**	[18,19,20,21]	[5]
GPx (U/mg protein)	5.02 ± 0.52	6.28 ± 1.23
GR (U/mg protein)	33.50 ± 9.70	31.24 ± 8.24
GSH (nmol/mg protein)	10.16 ± 2.10	12.24 ± 3.24
GSSG (nmol/mg protein)	1.85 ± 0.42	2.03 ± 0.54
GSSG/GSH	0.21 ± 0.09	0.23 ± 0.10
MDA (nmol/mg protein)	1.79 ± 0.52	1.68 ± 0.60

Note: Data are shown as the mean ± SD of several values corresponding to the numbers of subjects analyzed in each group (*n*). Chemotaxis index (C.I.); phagocytic index (*p*.I.); phytohemagglutinin (PHA) and lipopolysaccharide (LPS) lymphoproliferation in basal (c.p.m.) and stimulated (%) conditions; glutathione peroxidase (GPx) and glutathione reductase (GR) activities; intracellular reduced (GSH) and oxidized (GSSG) glutathione contents; GSSG/GSH ratios; and intracellular malondialdehyde (MDA) contents. Statistical analysis did not reveal significant differences between both groups of PD patients.

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
