# Peer review of "Lymphoproliferation Impairment and Oxidative Stress in Blood Cells from Early Parkinson’s Disease Patients"

_ijms, 2019, doi:10.3390/ijms20030771_

Reviewer 1 Report

This manuscript is to find peripheral biomarkers for PD patients at the early stage 2 comparing to age-matched controls, with a special focus on immune functions. The authors successfully demonstrated an early impairment of lymphoproliferation capacity and increased levels of oxidative stress and lipid peroxidation in WB cells of PD stage 2 patients. The reviewer thinks that the experimental design is straightforward, well-planned,  and the data were clearly presented, and the results are clean and significant. The findings are worth publishing and the contents are mostly reasonable. However, the length of introduction and also discussion sections are extremely long and redundant, poorly organized.

 Major comments

1.     Introduction needs to cut down to a half, without loosing important contents. First paragraph could be more organized and easily cut down to a half. Provide important and sufficient facts and basics to understand what authors are trying to do in this paper. No need to provide all the facts known in PD.  Since the authors did not analyze any neuronal damage or markers or changes in the CNS, the second paragraph (line 68 to 90) should be minimized, and just have a main focus to explain why the authors want to investigate functional profiling of peripheral immune cells.

 2.     Discussion should be cut down to two third. Needs more organization. Talking mostly about the same thing for the second paragraph (line 248 to line 281). I would like to know more about the mechanistic thoughts, why PD patients suffer an accelerated impairment of the acquired immunity, not innate functions.

 3.     Paragraph from line 308 to 347 and the following two paragraphs line 348 to 380 and line 381 and 393 are really overlapping. The same things appeared later in a similar context.  Actually, diluting the nice and clean data shown in this manuscript. The reviewer does not appreciate all these discussion without any visible points. Please prioritize and first talk about the main argument,  similar results or supporting results from other study, and brief conclusion from the authors’ side.

 Minor comments

 Sometimes, one sentence is too long, very difficult  to follow the logic. Probably, it needs to be professionally edited by a native English writer.

Author Response

REVIEWER 1

Major comments

1.Introduction needs to cut down to a half, without loosing important contents. First paragraph could be more organized and easily cut down to a half. Provide important and sufficient facts and basics to understand what authors are trying to do in this paper. No need to provide all the facts known in PD.  Since the authors did not analyze any neuronal damage or markers or changes in the CNS, the second paragraph (line 68 to 90) should be minimized, and just have a main focus to explain why the authors want to investigate functional profiling of peripheral immune cells.

     According to the suggestion of the Reviewer, the “Introduction” section has been shortened and reorganized, for which we have eliminated the information that seemed to us to be more irrelevant, such as neuronal damage or changes in the CNS (second paragraph).

2.Discussion should be cut down to two third. Needs more organization. Talking mostly about the same thing for the second paragraph (line 248 to line 281). I would like to know more about the mechanistic thoughts, why PD patients suffer an accelerated impairment of the acquired immunity, not innate functions.

 According to the suggestion of the Reviewer, the “Discussion” section has been shortened, for which we have summarized the main ideas, and we have also eliminated the information that seemed to us to be more irrelevant to discuss the results. Furthermore, we have included more information about the mechanistic thoughts, by why PD patients could suffer an accelerated impairment of the adaptive immunity.

3.Paragraph from line 308 to 347 and the following two paragraphs line 348 to 380 and line 381 and 393 are really overlapping. The same things appeared later in a similar context.  Actually, diluting the nice and clean data shown in this manuscript. The reviewer does not appreciate all these discussion without any visible points. Please prioritize and first talk about the main argument,  similar results or supporting results from other study, and brief conclusion from the authors’ side.

According to the suggestion of the Reviewer, the “Discussion” section have been reorganized and summarized, in order to clarify the main argument of the manuscript.

Minor comments

Sometimes, one sentence is too long, very difficult to follow the logic. Probably, it needs to be professionally edited by a native English writer.

Following the suggestion of the Reviewer, an extensive revision of the English language has been made throughout the entire manuscript in order to correct the style.

Reviewer 2 Report

This manuscript entitled “Lymphoproliferation impairment and oxidative stress in blood cells from early Parkinson’s disease patients” reports a set of experiments showing a decrease in adaptative inmmunity in PD patients but not in innate inmmunity response. The study is interesting and only minor points should be take into account.

- Both the Introduction and the Discussion sections are too long. I recommend the authors to summarize the main ideas.

- The authors should refer to figure 1F and figure 2G in the text.

- I recommend the author to include a figure with data from medicated and non-medicated patients.

Author Response

REVIEWER 2

The study is interesting and only minor points should be taken into account.

1. Both the Introduction and the Discussion sections are too long. I recommend the authors to summarize the main ideas.

According to the suggestion of the Reviewer, the “Introduction” and the “Discussion” sections have been shortened, for which we have summarized the main ideas, and we have also eliminated the information that seemed to us to be more irrelevant to discuss the results.

2. The authors should refer to figure 1F and figure 1G in the text.

According to the Reviewer, figure 1F and figure 1G have been referred in the corresponding text of the “Results” section.

3. I recommend the author to include a figure with data from medicated and non-medicated patients.

According to the Reviewer, a table (Table 1) with the data from medicated and non-medicated Parkinson Disease patients has been included in the corresponding text of the “Results” section, in order to facilitate the description of these results.